# System Integrated Information

**DOI:** 10.3390/e25020334

**Published:** 2023-02-11

**Authors:** William Marshall, Matteo Grasso, William G. P. Mayner, Alireza Zaeemzadeh, Leonardo S. Barbosa, Erick Chastain, Graham Findlay, Shuntaro Sasai, Larissa Albantakis, Giulio Tononi

**Affiliations:** 1Department of Psychiatry, University of Wisconsin-Madison, Madison, WI 53719, USA; 2Department of Mathematics and Statistics, Brock University, St. Catharines, ON L2S 3A1, Canada; 3Neuroscience Training Program, University of Wisconsin-Madison, Madison, WI 53705, USA; 4Fralin Biomedical Research Institute at VTC, Virginia Tech, Roanoke, VA 24016, USA; 5Department of Mathematics, University of Dallas, Irving, TX 75062, USA; 6Araya Inc., Tokyo 107-0052, Japan

**Keywords:** integrated information, consciousness, causation, intrinsic

## Abstract

Integrated information theory (IIT) starts from consciousness itself and identifies a set of properties (axioms) that are true of every conceivable experience. The axioms are translated into a set of postulates about the substrate of consciousness (called a complex), which are then used to formulate a mathematical framework for assessing both the quality and quantity of experience. The explanatory identity proposed by IIT is that an experience is identical to the cause–effect structure unfolded from a maximally irreducible substrate (a Φ-structure). In this work we introduce a definition for the integrated information of a system ($\varphi_{s}$) that is based on the existence, intrinsicality, information, and integration postulates of IIT. We explore how notions of determinism, degeneracy, and fault lines in the connectivity impact system-integrated information. We then demonstrate how the proposed measure identifies complexes as systems, the $\varphi_{s}$ of which is greater than the $\varphi_{s}$ of any overlapping candidate systems.

## 1. Introduction

Integrated information theory starts from the immediate and irrefutable fact that experience exists. The theory then identifies a set of five properties (axioms) that are irrefutably true of every experience: intrinsicality, information, integration, exclusion, and composition [1,2] (but see [3] for other properties that might be considered). The axioms are chosen with the understanding that they should be irrefutably true of every conceivable experience and complete (there are no other essential properties that characterize every experience) [4]. The zeroth axioms of IIT, **Existence**, states that experience *exists*: there is *something*. From there, the remaining axioms are defined as follows:**Intrinsicality** Experience is *intrinsic*: it exists *for itself*.**Information** Experience is *specific*: it is *the way it is*.**Integration** Experience is *unitary*: it is a *whole*, *irreducible* to separate experiences.**Exclusion** Experience is *definite*: it is *this* whole.**Composition** Experience is *structured*: it is composed of *distinctions* and the *relations* that bind them together, yielding a *phenomenal structure*.

The axioms are then translated into a set of postulates about the substrate of consciousness (called a complex) [5]. A substrate is a physical system, where physical is understood operationally as something that can be observed and manipulated. Just as the existence of experience is the basis for the remaining axioms, the postulates build off of a basic requirement for physical **Existence** (the zeroth postulate): to exist, the substrate of consciousness must have *cause–effect power*: there must be units that can *take and make a difference*. The remaining postulates are translations of the axioms in terms of cause–effect power:**Intrinsicality** The substrate of consciousness must have *intrinsic* cause–effect power: it must take and make a difference *within itself*.**Information** The substrate of consciousness must have *specific* cause–effect power: it must select a specific *cause–effect state*.**Integration** The substrate of consciousness must have *unitary* cause–effect power: it must specify its cause–effect state as *a whole* set of units, *irreducible* to separate subsets of units.**Exclusion** The substrate of consciousness must have *definite* cause–effect power: it must specify its cause–effect state as *this* set of units.**Composition** The substrate of consciousness must have *structured* cause–effect power: subsets of its units must specify cause–effect states over subsets of units (*distinctions*) that can overlap with one another (*relations*), yielding a *cause–effect structure*.

To assess whether (and if so, how) a substrate satisfies the postulates requires a mathematical framework for assessing cause–effect power [5]. In particular, the mathematical framework must define a measure of the irreducible cause–effect power of a system, the system integrated information ($\varphi_{s}$). System integrated information is based on the first four postulates (intrinsicality, information, integration, exclusion), and is used to determine whether or not a system is a complex (whether or not the cause–effect power of the system as a whole is maximally irreducible). The final postulate, composition, requires that we *unfold* in full the cause–effect power of the complex, yielding its cause–effect structure or Φ-structure, as presented in IIT 4.0 [3,6].

The mathematical formulation of IIT has been refined and developed over time, with the goal of fully and accurately capturing the postulates [5,7,8]. Recently, the mathematical framework has been updated by introducing a measure that uniquely satisfies the postulates of existence, intrinsicality, and information [6,9]. Moreover, there is now an explicit assessment of the causal relations among the distinctions specified by a system, as required by composition [10].

While IIT must still be considered a work in progress, in the following we introduce a definition of $\varphi_{s}$ that is consistent with these recent updates, is aligned with the postulates, and can be used to identify complexes. In Section 2, we introduce the mathematical framework for measuring $\varphi_{s}$, based on the first four postulates of IIT. In Section 3, we explore the behavior of the measure in several examples, and in Section 4, we summarize the results, discuss implications for IIT, and outline some future directions for research.

## 2. Theory

In this section we describe the mathematical framework used to assess the cause–effect power of a system. Our starting point is a stochastic system *U* with state space $\Omega_{U}$. Systematic interventions (manipulations) and measurements (observations) are used to define a transition probability function for the system: $$T_{U}\equiv p\left(\bar{u}|\mathrm{do}\left(u\right)\right),\;u,\bar{u}\in \Omega_{U},$$
which describes the probability of the observed state $\bar{u}$, given the intervention state *u*. It is assumed that the units of *U* are independent, given the current state of the system, so that: $$p\left(\bar{u}|\left(u\right)\right)=\prod_{i=1}^{\left|U\right|}p\left({\bar{u}}_{i}|\mathrm{do}\left(u\right)\right),\;u,\bar{u}\in \Omega_{U}.$$The elements of *U* are random variables that represent the units in the universal substrate (or, simply, the universe) under consideration, and $u\in \Omega_{U}$ is the current state of the universe. The goal is to define the system-integrated information ($\varphi_{s}$) of a system in a state S=s⊆U=u, based on the postulates of IIT, and use it to identify complexes.

### 2.1. Intrinsicality

For a system *S* to exist, it must be possible for something to change its state, and it must be able to change the state of something; in other words, it must have cause–effect power. For a system to exist intrinsically, it must have cause–effect power within itself. Thus, we consider the cause–effect power the system S=s⊆U=u has over itself. To this end, we *causally condition* all units outside of the system (W=U∖S) in their current state (*w*), which are considered as *background conditions*, and only consider the conditional transition probabilities:(1)$$T_{S}\equiv p\left(\bar{s}|\mathrm{do}\left(s\right)\right)=p\left(\bar{s}|\mathrm{do}\left(s\right),\mathrm{do}\left(w\right)\right),\;s,\bar{s}\in \Omega_{S},$$
throughout the causal analysis.

### 2.2. Information

The information postulate states that a system must select a specific cause–effect state. To identify the specific cause–effect state, we quantify the intrinsic information that the system, in its current state *s*, specifies about each possible cause or effect state $\bar{s}\in \Omega_{S}$.

Intrinsic information is defined as a product of two terms, *informativeness* and *selectivity* [6,9]. The informativeness term captures cause–effect power by measuring how being in state *s* increases the probability of an effect state, or how a cause state increases the probability of being in state *s*, relative to chance (and measured on a log scale). The selectivity term captures the specificity of the cause–effect power by quantifying how it is concentrated over a specific state (at the expense of other states).

The intrinsic effect information is defined as:$$\begin{array}{cc} {\mathrm{ii}}_{e}\left(s,\bar{s}\right) & =p_{e}\left(\bar{s}|s\right)\log\left(\frac{p_{e}\left(\bar{s}|s\right)}{p_{e}\left(\bar{s}\right)}\right), \end{array}$$
where the $p_{e}$s are the effect repertoires of the system. The constrained repertoire $p_{e}\left(\bar{s}|s\right)$ is a conditional probability distribution derived from Equation (Equation 1) that describes how the system *S*, by being in state *s*, constrains its potential effect state $\bar{s}\in \Omega$, and the unconstrained repertoire $p_{e}\left(\bar{s}\right)$ is the marginal distribution of effect states arising from a uniform distribution of intervention states (see Appendix A). The use of a uniform distribution of intervention states is required to accurately capture its full cause–effect power [11,12].

The intrinsic cause information is defined as:$$\begin{array}{cc} {\mathrm{ii}}_{c}\left(s,\bar{s}\right) & =p_{c}\left(\bar{s}|s\right)\log\left(\frac{p_{e}\left(s|\bar{s}\right)}{p_{e}\left(s\right)}\right), \end{array}$$
where the $p_{c}$ is the corresponding cause repertoire of the system. The cause repertoire inverts the effect repertoire (using Bayes’ Theorem), based on a uniform marginal distribution of the intervention state (see Appendix A). Note that for both the cause and effect intrinsic information, the informativeness term uses the effect repertoire to capture how the system increases the probability of a state. However, for ${\mathrm{ii}}_{e}$ it is the probability of the effect state given the current state of the system, while for ${\mathrm{ii}}_{c}$ is the probability of the current state of the system given the cause state.

Together, informativeness and selectivity define a measure that is sub-additive [9]. When the system is fully selective ($p_{c/e}\left(\bar{s}|s\right)=1$), the measure is additive over units, meaning the cause–effect power over the system is equal to the sum of the cause–effect power over individual units (expansion). However, when the cause–effect power of the system is not fully selective ($p_{c/e}\left(\bar{s}|s\right)<1$), the cause–effect power of the system is less than the sum of the cause–effect power over individual units (dilution). This occurs because the cause–effect power is spread over multiple possible states, yet the system must select one.

The intrinsic information of a system about a state $\bar{s}$ is a measure of its specific cause–effect power, and, as such, a measure of specific existence. To identify the specific state selected by the system, IIT appeals to the principle of maximal existence [3]. The principle states that, with respect to essential requirements for existence, what exists is what exists the most. Accordingly, the cause and effect states specified by a complex are defined as the ones that maximize the intrinsic cause and effect information,
$$\begin{array}{cc} s_{c/e}^{\prime }= & \underset{\bar{s}\in \Omega_{S}}{\mathrm{argmax}}{\mathrm{ii}}_{c/e}\left(s,\bar{s}\right) \end{array}$$

### 2.3. Integration

The integration postulate states that the cause–effect power of a complex must be unitary, which means it must specify its cause–effect state as a whole set of units, *irreducible* to separate subsets of units. From the information postulate, we have already identified the cause–effect state specified by the system ($s_{c}^{\prime }$ and $s_{e}^{\prime }$). To evaluate irreducibility, we ask whether the system specifies its cause–effect state in a way that is irreducible to separate parts. To do so, we cut the system into parts using directional partitions (see Appendix A), and measure the difference the cut makes to the intrinsic information specified by the system over its cause–effect state.

A directional partition is a partition of the system *S* into K≥2 parts, such that each part has either its inputs, outputs, or both, cut away from the rest of the system. A partition θ of *S* has the form:$$\theta =\{S_{\delta_{1}}^{\left(1\right)},S_{\delta_{2}}^{\left(2\right)},\ldots ,S_{\delta_{K}}^{\left(K\right)}\},$$
where $\left\{S^{\left(i\right)}\right\}$ is a partition of *S*,
$$S^{\left(i\right)}\neq ⌀,\;S^{\left(i\right)}\cap S^{\left(j\right)}=⌀,\;\overset{K}{\underset{i=1}{⋃}}S^{\left(i\right)}=S,$$
and each $\delta_{i}\in \left\{\leftarrow ,\to ,↔\right\}$ indicates whether its inputs (←), outputs (→), or both (↔), are cut. For a system *S*, we define the set of all possible directional partitions as Θ(S).

Together, integration and existence require that, for every possible part of the system, the rest of the system both makes a difference to it (produces an effect) and takes a difference from it (bears a cause). To assess this, we need directional partitions (see Appendix A for an example).

Given a partition θ∈Θ(S), we define a partitioned transition probability function ($T_{S}^{\theta }$) and corresponding partitioned effect repertoires ($p_{e}^{\theta }$) to describe the cause–effect power that remains after cutting the indicated inputs and outputs (see Appendix A). The partitioned effect repertoires describe the probability of an effect state given a current state (or the probability of a current state given a cause state) after the connections between parts have been cut. The integrated cause or effect information of the system *S* over the partition θ, is defined as the intrinsic cause or effect information of *S*, with informativeness defined relative to the partitioned repertoire, instead of the unconstrained repertoire. Analogous to intrinsic information, the integrated effect information is defined as:$$\varphi_{e}\left(s,\theta \right)=p_{e}\left(s_{e}^{\prime }|s\right)\log{\left(\frac{p_{e}\left(s_{e}^{\prime }|s\right)}{p_{e}^{\theta }\left(s_{e}^{\prime }|s\right)}\right)}_{+},$$
and the integrated cause information is:$$\varphi_{c}\left(s,\theta \right)=p_{c}\left(s_{c}^{\prime }|s\right)\log{\left(\frac{p_{e}\left(s|s_{c}^{\prime }\right)}{p_{e}^{\theta }\left(s|s_{c}^{\prime }\right)}\right)}_{+},$$
where the + subscript indicates the positive part (negative values are set to zero).

The integrated information of a system is a measure of its irreducible, specific cause–effect power, and, as such, a measure of irreducible, specific existence. To quantify the irreducible cause–effect power of a system, IIT appeals to the principle of minimal existence [3], which complements the principle of maximal existence. The principle of minimal existence states that, with respect to an essential requirement for existence, nothing exists more than the least it exists. Accordingly, since a system must both take and make a difference to exist, the system integrated information for a given partition θ is defined as the minimum of its integrated cause and effect information,
$$\varphi_{s}\left(s,\theta \right)=\min\left\{\varphi_{c}\left(s,\theta \right),\varphi_{e}\left(s,\theta \right)\right\}.$$

Moreover, there are many ways to cut a system into separate parts. By the principle of minimal existence, a system cannot exist as *one* system more than it exists across its weakest link. Accordingly, we define the integrated information to be the intrinsic information of the system relative to its minimum partition $\theta^{\prime }$,
$$\varphi_{s}\left(s\right)=\varphi \left(s,\theta^{\prime }\right).$$

The minimum partition is the “weakest link” or *fault line* of the system, defined as the partition θ∈Θ(S) that minimizes the integrated information relative to the maximum possible integrated information for the given partition,
$$\theta^{\prime }=\underset{\theta \in \Theta (S)}{\mathrm{argmin}}\frac{\varphi_{s}\left(s,\theta \right)}{\max_{T_{S}}\varphi_{s}\left(s,\theta \right)}.$$

Relative integrated information quantifies integration as the strength of the connections among parts. It identifies the minimum partition of the system in a way that does not depend on the number of parts and their sizes. By contrast, the integrated information of the system is an absolute quantity, quantifying the loss of intrinsic information due to the minimum partition of the system. The use of relative integrated information to define the minimum partition explains why we must consider cutting both the inputs and outputs of a part, as well as why we need to partition the system into K≥2 parts. Although such partitions cut “more” in an absolute sense, they may be identified as the minimum partition if the increase in integration information is outpaced by the increase in maximum possible integrated information, resulting in an overall decreased integration.

For a given θ∈Θ(S), the maximum possible value of $\varphi_{s}\left(s,\theta \right)$ is equal to the number of potential connections cut by the partition, as demonstrated in the following theorem:

**Theorem** **1.**
*Let Θ(S) be the set of directional partitions of a system S, and let θ∈Θ(S) be any directional partition. For each $S^{\left(i\right)}\in \theta$, define $X^{\left(i\right)}$ to be the set of units whose output to $S^{\left(i\right)}$ has been cut by the partition. The maximum possible value of φ(s,θ) is*

$$\max_{\left(T_{S}\right)}\varphi_{s}\left(s,\theta \right)=\sum_{i=1}^{K}|S^{\left(i\right)}\left|\right|X^{\left(i\right)}|$$



See proof in Appendix B.

### 2.4. Exclusion

The exclusion postulate states that the cause–effect power of a complex must be definite, in that the complex must specify its cause–effect state as a definite set of units (with a definite spatiotemporal grain). The question is, which set? In general, multiple candidate systems with overlapping units may have positive values of $\varphi_{s}$. Based again on the principle of maximal existence, we pick the one that exists the most; in this case the one that exists the most as an irreducible substrate. Accordingly, we identify a complex by finding the maximally irreducible system $S^{*}=s^{*}\subseteq U=u$:$$S^{*}=\underset{S=s\subseteq U=u}{\mathrm{argmax}}\varphi_{s}\left(s\right)$$
with corresponding selected cause and effect states:$$s_{c/e}^{*}=\underset{\bar{s}\in \Omega_{S^{*}}}{\mathrm{argmax}}{\mathrm{ii}}_{c/e}\left(s^{*},\bar{s}\right).$$In principle, the search should include not only subsets of U=u, but also include systems of units with different spatiotemporal grains. For simplicity, in this work we restrict consideration to the grain at which the universe is defined.

The maximally irreducible system $S^{*}$ is called a complex within *U*. Any systems overlapping with $S^{*}$ are then excluded from further consideration. The process of identifying complexes can then be applied recursively to carve *U* into non-overlapping maximal substrates. It is possible that two or more overlapping systems tie as maximally irreducible. In this situation, neither system is considered definite and the systems exclude each other. The process continues with the two systems removed from consideration. A recursive algorithm for identifying complexes is presented in Appendix C. Note that, in addition to $\varphi_{s}$, there are other aspects of the mathematical framework that could result in a tie. If two or more cause–effect states are tied for having the maximal intrinsic information, then, as an extension of the principle of maximal existence, we take the one that maximizes $\varphi_{s}$. A similar approach is applied if there are two or more partitions tied for the minimum partition. However, if the two states (or two partitions) lead to equal values of $\varphi_{s}$, then it is not necessary to resolve the tie in order to identify complexes.

## 3. Results and Discussion

In this section we present $\varphi_{s}$ for several small systems to highlight properties that influence system-integrated information. All computations were performed using the PyPhi package for computing integrated information [13]. The examples focus on $\varphi_{s}$ and identifying complexes (and not on the resulting Φ-structures, but see [3]). The first set of examples explored the role of the intrinsic information of a system (information), the second set of examples explored the role of integrated information (integration), and the third set of examples explored how these aspects combine to yield a maximum of system-integrated information (exclusion).

### 3.1. Example 1: Information

The first set of examples focused on two factors that influence intrinsic information: determinism and degeneracy [14,15,16]. A system in a state is called deterministic when it specifies an effect state with probability one. By contrast, a system in a state is nondeterministic if there are multiple potential effect states with non-zero probability. A system in a state is called non-degenerate if it specifies a cause state with probability one. By contrast, a system in a state is degenerate if there are multiple potential cause states with non-zero probability in the cause repertoire.

First, we considered a system of four interconnected, deterministic units, each with a unique input–output function (Figure 1A). The current state of the system was that units {A,B,D} were ON, and unit {C} was OFF (indicated by the black circles and upper-case labels vs. white circles with lower case labels). The system specified a unique cause–effect state (Figure 1B) and, as a result, had high intrinsic cause and effect information (${\mathrm{ii}}_{c}={\mathrm{ii}}_{e}=4$).

To highlight the impact of indeterminism, we considered a modified system where unit D was noisy (Figure 1C). Given the current state of the system, unit D went into the effect state specified by the system in Figure 1A with probability 0.6, and it went into the opposite state with probability 0.4. The indeterminism in unit D reduced the intrinsic information of the system relative to the deterministic system (${\mathrm{ii}}_{c}=1.95$, ${\mathrm{ii}}_{e}=1.95$).

To highlight the impact of degeneracy, we considered a modified system where the input–output function of unit D was identical to that of unit A (Figure 1D). In this case, the system was still deterministic, but unit D led to degeneracy in the system, as there were now two cause states that could lead to the current state of the system, both with equal intrinsic information. The intrinsic cause information of these states was reduced, compared to the nondegenerate system, because the selectivity of the cause states decreased and the unconstrained probability of the current state increased (${\mathrm{ii}}_{c}=1.5$). (There were two states tied for the maximum intrinsic information; however, for this system both states led to the same value of $\varphi_{s}$). The units in this example were still deterministic, but the intrinsic effect information also reduced, because the unconstrained probability of the effect state increased (${\mathrm{ii}}_{e}=3.0$).

Although determinism is defined by the selectivity of effect states, and degeneracy is defined by the selectivity of cause states, both can have an impact on the intrinsic cause and effect information. In general, higher values of intrinsic information allow for higher values of integrated information. Thus, high determinism and low degeneracy are necessary conditions for high $\varphi_{s}$, although they are not sufficient.

While the IIT formalism evaluates the intrinsic information the system specifies about itself, in adaptive, open systems, environmental factors shape the intrinsic interactions within the system [17,18,19]. Which environmental constraints facilitate the evolution of high $\varphi_{s}$ is an important topic for future research (but see [17,20,21] for prior versions of IIT).

### 3.2. Example 2: Integration

The second set of examples focused on the connectivity of the system, and how fault lines in the system reduced integration. For these examples, the universe *U* was constituted of four units $U_{i}$, with state space $\Omega_{U_{i}}=\left\{-1,+1\right\}$, each with a sigmoidal activation function:(2)$$p\left(+1|u\right)=\frac{1}{l+\exp\left(-k\sum_{j=1}^{n}w_{ji}u_{j}\right)},$$
where
$$\sum_{j=1}^{n}w_{ji}=1\;\forall \;i.$$The units could be thought of as noisy threshold units, where each $w_{ji}$ defined the contribution of unit *j* to the activation function of unit *i*, *k* was a noise (determinism) parameter that defined the slope of the sigmoid, and *l* was a bias for the unit to be ON ($u_{i}=+1$) or OFF ($u_{i}=-1$). For these examples we used the value k=3 (moderate noise) and l=1 (unbiased). For simplicity, we considered systems in the all OFF state (white circles, lower case label). We explored the impact of connectivity on $\varphi_{s}$ by looking at systems with different weights $w_{ji}$. To highlight the impact of fault lines per se (as opposed to some combination of fault lines, indeterminism, and degeneracy), for each example we reported the integrated information relative to the intrinsic information of the system.

First, we considered a system of four units (denoted A, B, C, D) with a cycle of strong connections (w=0.4) and a reverse cycle of slightly weaker connections (w=0.3), a weak self-loop (w=0.2), and even weaker connections to non-neighboring units (w=0.1). The connectivity structure was symmetric, and there were no fault lines in the system (see Figure 2A). For this system, there were two equivalent minimum partitions, either $\theta^{\prime }=\left\{{\left\{AB\right\}}_{↔},{\left\{CD\right\}}_{↔}\}\right\}$ or $\theta^{\prime }=\left\{{\left\{AD\right\}}_{↔},{\left\{BC\right\}}_{↔}\}\right\}$, and the resulting integrated information was $\varphi_{s}=0.3393$ for both. For this system, the integrated cause and effect information were 48.1% of the intrinsic cause and effect information. Note that, in general, the proportions are not necessarily the same. However, for these examples they were the same because the state of the current state of the system was the same as the selected cause and effect states.

Second, we considered a system where three units {A,B,C} had strong, bidirectional connections with each other (w=0.3), weaker self-connections and bidirectional connections with *D* (w=0.2). The remaining unit *D* had a strong self-connection (w=0.4) and weak bidirectional connections with the other units (w=0.2). For this system, there was a fault line between {A,B,C} and {D}, which was identified by $\theta^{\prime }=\left\{{\left\{ABC\right\}}_{\leftarrow },{\left\{D\right\}}_{\to }\right\}$ (see Figure 2B). The resulting integrated information was $\varphi_{s}=0.0628$, and the integrated cause and effect information was 10.0% of the intrinsic cause and effect information.

In the third system, the weights were selected such that the four units formed two strongly connected pairs {A,B} and {C,D}, with strong connections within pairs (w=0.4), weak connections between pairs (w=0.15), and moderate strength self-connections (w=0.3). For this system there was a fault line between {A,B} and {C,D}, which was identified by $\theta^{\prime }=\left\{{\left\{AB\right\}}_{↔},{\left\{CD\right\}}_{↔}\right\}$ (see Figure 2C). The resulting integrated information was $\varphi_{s}=0.1477$, the integrated cause and effect information was 21.2% of the intrinsic cause and effect information.

Thus, while higher values of intrinsic information enabled higher values of integrated information, when there were fault lines in the system, integrated information was a lower proportion of the intrinsic information.

### 3.3. Example 3: Exclusion

The final example demonstrated exclusion, and how a universal substrate could be condensed into non-overlapping systems with maximally irreducible cause–effect power. For this example, we used a universe *U* of eight units, where each $U_{i}$ had the sigmoidal activation function described in Equation (Equation 2) (l=1). Six of the units, {A,B,C,D,E,F}, had a moderate level of noise (k=2), and the last two, {G,H}, had a high level of noise (k=0.2).

Within the universe, there was a cluster of 5 units {A,B,C,D,E} with the following connectivity pattern: weak self-connection (w=0.025), strong input from one unit within the cluster (w=0.45), moderate input from one unit within the cluster (w=0.225) and weak input from the other two units in the cluster (w=0.1). The weights were arranged such that the connectivity pattern of the 5-unit cluster was symmetrical, with the strong connections forming a loop (A→B→C→D→E→A; no fault lines; see Figure 3A). The 5-unit cluster had weak, bidirectional connections with the three units outside the cluster (w=0.033). For the three units not in the cluster {F,G,H}, unit *F* had a strong self-connection (w=0.769) and weak inputs from *G* and *H* (w=0.033), while *G* and *H* had a strong bidirectional connection (w=0.769) and weak self-connections and input from *F* (w=0.033).

To define complexes within the universe *U*, we computed the system integrated information for every candidate system S⊆U to find $S^{*}$, the system that maximized $\varphi_{s}$ (Figure 3B). Any systems that overlapped with $S^{*}$ were then excluded from future consideration, and the process was applied recursively, according to the algorithm in Appendix C. For this universe, the process resulted in three complexes, {F} with $\varphi_{s}=0.49$, {A,B,C,D,E} with $\varphi_{s}=0.12$, and {G,H} with $\varphi_{s}=0.06$ (see Figure 3C). In this universe, there were multiple systems with $\varphi_{s}>0$ that did not constitute a complex, because they overlapped a complex with higher $\varphi_{s}$ (Figure 3D).

To understand why the universal substrate condensed in this way, we considered a nested sequence of systems {A},{A,B},{A,B,C},{A,B,C,D},{A,B,C,D,E}. Starting from {A}, each time a new unit was added to the candidate system, both the intrinsic cause and effect information increased (see Figure 3E). However, the $\varphi_{s}$ values were consistently low for the first four systems in the sequence, because there was always a fault line in the system, and only increased substantially for the five units (see Figure 3F). By cutting the inputs to the last unit added to the system, we could avoid cutting any strong connections (with w=0.45). When the fifth unit {E} was added, the system connectivity was symmetric and there were no longer any fault lines, resulting in an increase in the $\varphi_{s}$ value. On the other hand, adding any of {F,G,H} to the 5-unit system introduced a new fault line in the system and resulted in decreased $\varphi_{s}$.

High determinism, low degeneracy, and a lack of fault lines are properties that allow systems to have high $\varphi_{s}$. However, for defining a complex, it is not the absolute value of $\varphi_{s}$ that matters, only whether or not it is maximal. Nevertheless, ‘local’ maxima of $\varphi_{s}$, while not substrates of consciousness, may still reveal biologically, or functionally, interesting systems [22,23]. The algorithm in Appendix C could be used to carve a universe into non-overlapping maximally irreducible substrates.

## 4. Conclusions

This work introduces a definition of system integrated information that closely tracks the postulates of IIT in its latest formulation [3]. The main goal of IIT 4.0, compared to previous formulations, is to strengthen the link between the axioms, postulates, and the mathematical framework. The definition of system integrated information relies on a recently introduced measure of intrinsic information, which uniquely satisfies the postulates of existence, intrinsicality, and information [9]. The irreducibility of a system, measured by system integrated information, is related to the measure of integrated information for mechanisms within a system, introduced in [6].

In this paper, we employed minimalistic causal models to illustrate the key properties of substrates that influence integrated information and, ultimately, determine whether a substrate is a complex. We first demonstrated that the intrinsic information of a system, and, thus, its potential for integrated information, is influenced by indeterminism and degeneracy [16]. We, then, demonstrated how systems with fault lines in their connectivity structure have reduced integrated information, because a lower proportion of their intrinsic information is integrated [23]. Finally, we explored how these aspects (indeterminism, degeneracy, fault lines) influence whether a system is a complex (has maximal $\varphi_{s}$), and how a universal substrate condenses recursively into non-overlapping complexes.

This paper formalizes mathematically how a substrate of consciousness can be identified, based on the postulates of existence, intrinsicality, information, integration and exclusion. Once a complex has been identified, its cause–effect power must be *unfolded* in full, as required by the postulate of composition, to yield its Φ-structure, as described in [3]. According to IIT’s explanatory identity, the properties of the Φ-structure unfolded from a complex fully account for the specific quality of the experience supported by the complex in its current state.

Forthcoming work will illustrate how bounds for $\varphi_{s}$ can be derived and will explore the relationship between $\varphi_{c/e}$ and ${\mathrm{ii}}_{c/e}$. Future work will also investigate the properties of more realistic networks and establish which aspects of their overall connectivity patterns, local specialization, and intrinsic mechanisms of macro units, allow them to constitute large complexes. This will provide a solid theoretical foundation for testing several predictions of IIT; the first among them being the prediction that the overall maximum of system integrated information in the human brain should correspond to the substrate of consciousness indicated by clinical and experimental evidence.

## Figures and Tables

**Figure 1 entropy-25-00334-f001:**
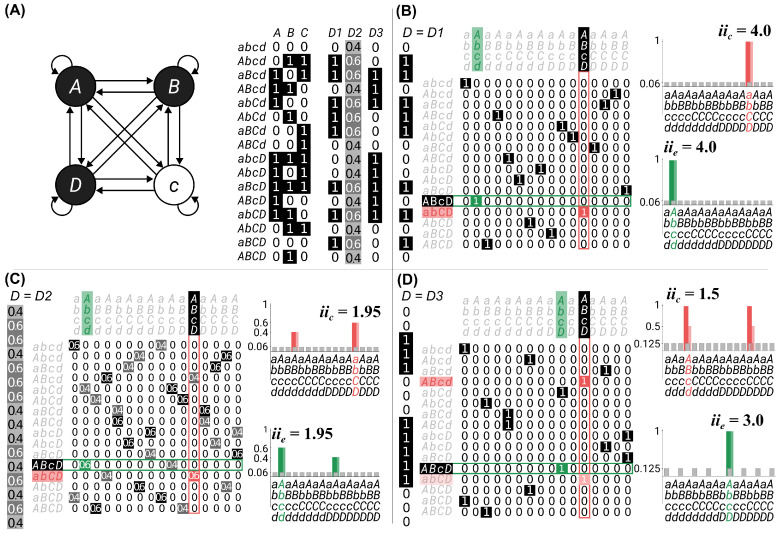
A set of example systems used to explore the impact of indeterminism and degeneracy on system intrinsic information. Dark colored bars and grey bars to the right of the transition probabilities in (**B**–**D**) represent the quantities for informativeness (constrained and unconstrained) and light colored bars for selectivity. (**A**) A base system constituted of four units. The units were all-to-all connected, and each unit had a unique input–output function, described in the state-by-node transition probability matrix (each column defines the probability a unit is ON, given the previous state defined by the row). Black circles and capital letters indicate the current state is ON, while white circles with lower case letters indicate the current state is OFF. (**B**) The state-by-state transition probability matrix ($T_{S}$) of the system in (**A**), and the corresponding cause and effect repertoires used to identify $s_{c/e}^{\prime }$. The system was deterministic (one non-zero value in each row), non-degenerate (one non-zero value in each column), and had high values of intrinsic cause and effect information $\left({\mathrm{ii}}_{c}={\mathrm{ii}}_{e}=4\right)$. (**C**) The same system as in (**B**), but with unit D noisy, so that it went into the state specified by system (**B**) with probability 0.6, and the opposite state with probability 0.4. The system was now non-deterministic (more than one non-zero entry in each row) and degenerate (more than one non-zero entry in each column), and the cause and effect intrinsic information decreased $\left({\mathrm{ii}}_{c}=1.95,{\mathrm{ii}}_{e}=1.95\right)$. (**D**) The same system as in (**B**), but unit D was changed so that it’s input–output function was identical to that of unit A. The system was deterministic (one non-zero entry in each row) but degenerate (more than one non-zero entry in each column), and had reduced cause and effect intrinsic information (${\mathrm{ii}}_{c}=1.5,{\mathrm{ii}}_{e}=3$).

**Figure 2 entropy-25-00334-f002:**
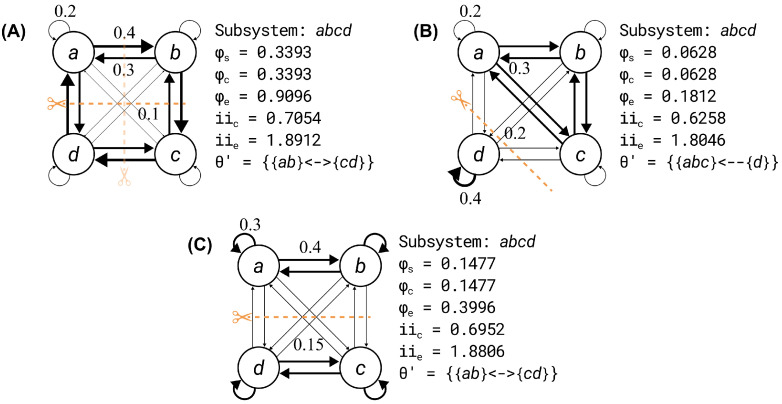
A set of example systems used to explore the impact of connectivity on system integrated information. In this example, the systems were constituted of four units in the OFF state, each with a sigmoidal activation function (Equation (Equation 2); k=3,l=1) and varying weights. The minimum partition for each system is indicated by the dashed orange line(s). (**A**) The first system had a symmetric connectivity structure with no fault lines. Two equivalent minimum partitions were identified, cutting {AB} away from {CD} or {AD} away from {BC}. The integrated information was 48.1% of the intrinsic information. (**B**) A system with 3 strongly interconnected units {A,B,C}, and a fourth unit {D} that was weakly connected with the rest. A directional partition of {ABC} from {D} was the minimum partition of the system, and, as a result, the integrated information was only 10.0% of the intrinsic information. (**C**) A system with 2 strongly interconnected sets of units {AB} and {CD} with weak connections between them. Cutting {AB} away from {CD} (bidirectionally) was the minimum partition of the system, and the resulting integrated information was only 21.2% of the intrinsic information.

**Figure 3 entropy-25-00334-f003:**
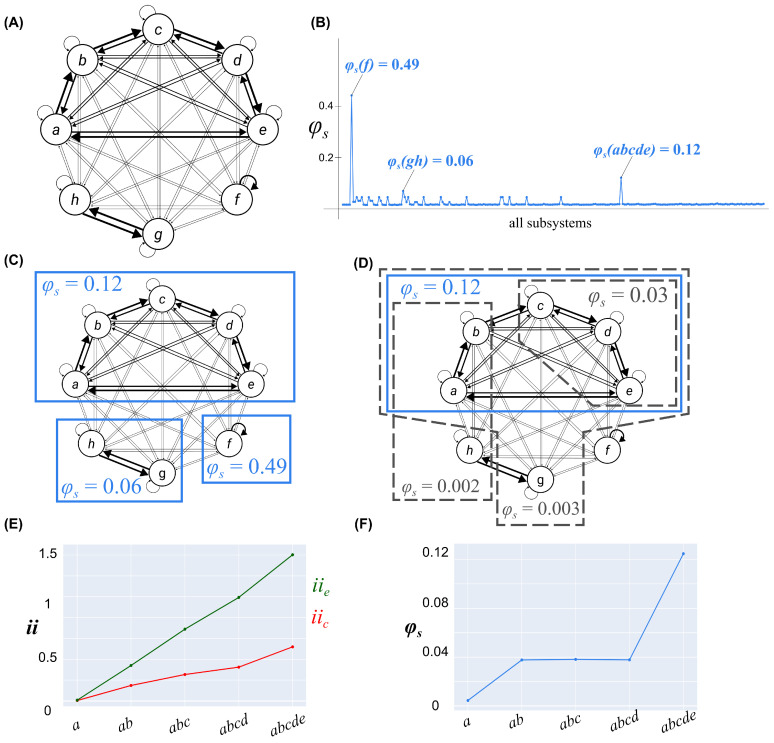
A universal substrate used to explore how degeneracy, determinism, and fault lines impacted how it condensed into non-overlapping complexes. (**A**) A universal substrate of 8 units, each having a sigmoidal activation function (Equation (Equation 2) (l=1). Units {A,B,C,D,E,F} has a moderate level of determinsism (k=2) and units {G,H} had a low level of determinism (k=0.2). A cluster of 5 units {A,B,C,D,E} had stronger (but varying) connections within the cluster than without, a unit {F} had a strong self-connection and weak connections with everything else, and two units {G,H} were strongly connected between themselves with weak self-connections and weak connections to the rest of the units. (**B**) The system integrated information values for the potential systems S⊆U. (**C**) The universal substrate condensed into three non-overlapping complexes, {A,B,C,D,E}, {F}, and {G,H}, according to the algorithm in Appendix C. A solid blue line around a set of units indicates that it constituted a complex. (**D**) A sample of potential systems (dashed grey outline) that were excluded because they were a subset of a complex ({C,D,E}), a superset of a complex ({A,B,C,D,E,G}), or a “paraset”, partially overlapping a complex ({A,B,H}). Each of these candidate systems had lower $\varphi_{s}$ than {A,B,C,D,E}. (**E**) The intrinsic cause and effect information of a nested sequence of candidate systems. (**F**) The integrated information of the systems in (**E**). The intrinsic information increased with each new unit added to the candidate system, but only when all five units were considered were there no fault lines in the system, leading to a maximum of integrated information.

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
