# Peer review of "System Integrated Information"

_entropy, 2023, doi:10.3390/e25020334_

Round 1

Reviewer 1 Report

This article is a companion piece to the main article on integrated information theory (IIT) version 4.0, that has just appeared on arxiv. That other article presents the whole theory, with a focus on the integrated information cause-effect structures that supposedly correspond to conscious contents (qualia) associated with physical states. This paper provides extra detail on the system integrated information measure that quantifies conscious level, and which supposedly determines the set of system components that contribute to the qualia of the system at any given moment. The manuscript describes this new version of system integrated information clearly, and by way of examples, shows some of its properties: a system for which the current state determines well the future state (deterministic) and the past state (non-degenerate), and one which is well-connected (amongst its components) tends to have the highest system integrated information.

As with previous versions of IIT, version 4.0, will attract plenty of lively discussion, and so the manuscript will reach a sizeable audience. There’s plenty I could say about the pros and cons of the theory, but I don’t consider here to be the place to do so. I think the manuscript can be published more or less as is. My only suggestion would be to increase the font size in figure 1- parts of that are too small to be readable on a hard copy.

Author Response

Thank you for the positive assessment of the manuscript. We have modified Figure 1 as requested. 

Reviewer 2 Report

It is deeply exciting to read about these developments in a research program that draws together some of the most interesting topics one can contemplate: How do systems exist as wholes that are greater than the sum of their parts? What is consciousness? What is free will?

While I believe the paper could be published as is, I would like to see the following issues addressed (even if briefly):

Is IIT 4.0 considered to be the final iteration of IIT? Why or why not?

It has been argued that spatial, temporal, and causal coherence for modeled entities may be irrefutably true of every conceivable experience, and potentially a necessary part of compositionality (i.e., an experienced world where different stuff has different properties (the term “stuff” is used precisely here, in order to be distinguished from a more stringent criteria of experiencing “things”/objects)):

Frontiers | Integrated world modeling theory expanded: Implications for the future of consciousness (frontiersin.org)

The possibility of additional axioms should be briefly mentioned, even if highly contested. This is a crucial issue, in that only a jointly sufficient set would allow IIT to be used to infer consciousness based on the degree of integrated information in a system. As such, the introduction of “intrinsic information” as a unique mapping is highly notable.

This work is also relevant in that it addresses the issue of connecting IIT to other formalisms, such as the Free Energy Principle and Active Inference framework (FEP-AI). An IIT theorist would likely respond that this kind of functionalism is unhelpful in that it misses the point of starting from experience and proceeding axiomatically. However, even if IIT does not require such a functional grounding, it may nonetheless be useful for enriching other approaches, such as the relational ontology of FEP-AI, or ‘global’ neuronal workspace theories. And while perhaps not necessary for IIT’s theoretical validity, such connections to other frameworks may be useful for avoiding some confusions. In the paper linked above, computational ideas are used to show how the “expander graph critique” may actually constitute compelling evidence in favor of IIT (in addition to having correspondences with the phenomenology of space). Such functional correspondences may also help with suggesting Phi approximation methods, as suggested in the appendix of the aforementioned paper.

In mentioning the zeroth axiom of existence, it might be worth briefly mentioning Descartes so interested readers can be pointed to sources of inspiration for the IIT approach with its rich philosophical underpinnings (even if IIT could be thought of as a new beginning for the rational approach to philosophy).

While the principle of minimal existence seems irrefutable, the principle of maximal existence seems more questionable. Could there be a conflation of irreducible-cause-effect-power with physical conservation laws, or are these proposed to be one and the same?

Are there any advantages to relaxing the Exclusion axiom to allow IIT to serve as a more flexible model of causal emergence (e.g. applying it to multi-layer networks, such as those identified by edge-centric time series methods in fMRI?), and potentially allow it to more effectively interface with functional theories of consciousness? Note: It is understood that none of the axioms can be relaxed without compromising the reliability of starting from essential properties of experience and then postulating a sufficient set of mechanistic realizers. The suggestion here is that IIT could also have broader realizability apart from consciousness, as in these papers:

Integrated information as a common signature of dynamical and information-processing complexity: Chaos: An Interdisciplinary Journal of Nonlinear Science: Vol 32, No 1 (scitation.org)

Frontiers | An Integrated World Modeling Theory (IWMT) of Consciousness: Combining Integrated Information and Global Neuronal Workspace Theories With the Free Energy Principle and Active Inference Framework; Toward Solving the Hard Problem and Characterizing Agentic Causation (frontiersin.org)

What kinds of functional relationships exist between determinism and degeneracy in real systems? Reliable directed functioning vs. evolvability?

The paper ends in gesturing towards future work on realistic networks. While unnecessary for this publication, for which a more focused technical handling seems most appropriate, at some point I would suggest applying IIT’s analyses to intuitive interpretable systems, even if some precision is lost in the process. I believe this will increase people’s willingness to learn about this deeply profound (and dare I say it, beautiful) theory.

Round 2

Reviewer 2 Report

The authors have thoughtfully and thoroughly addressed all of the issues I raised.

Looking forward to seeing what comes next.